# Phase transitions associated with magnetic-field induced topological orbital momenta in a non-collinear antiferromagnet

Sihao Deng [1,2,3] ✉, Olena Gomonay [4], Jie Chen[1,3], Gerda Fischer [2], Lunhua He [3,5,6] ✉, Cong Wang[7], Qingzhen Huang[8], Feiran Shen[1,3], Zhijian Tan[1,3], Rui Zhou [5], Ze Hu[9], Libor Šmejkal [4], Jairo Sinova[4], Wolfgang Wernsdorfer [2,10] & Christoph Sürgers [2] ✉

Resistivity measurements are widely exploited to uncover electronic excitations and phase transitions in metallic solids. While single crystals are preferably studied to explore crystalline anisotropies, these usually cancel out in polycrystalline materials. Here we show that in polycrystalline $Mn_3Zn_{0.5}Ge_{0.5}N$ with non-collinear antiferromagnetic order, changes in the diagonal and, rather unexpected, off-diagonal components of the resistivity tensor occur at low temperatures indicating subtle transitions between magnetic phases of different symmetry. This is supported by neutron scattering and explained within a phenomenological model which suggests that the phase transitions in magnetic field are associated with field induced topological orbital momenta. The fact that we observe transitions between spin phases in a polycrystal, where effects of crystalline anisotropy are cancelled suggests that they are only controlled by exchange interactions. The observation of an off-diagonal resistivity extends the possibilities for realising antiferromagnetic spintronics with polycrystalline materials.

The combined approach of experimental and theoretical research has recently led to a deeper understanding of electronic transport phenomena in antiferromagnetic (AFM) materials, fostering the development of future antiferromagnetic spintronic devices for which the manipulation of antiferromagnetic order by external means is a key issue[1–6]. In particular, antiferromagnets with non-collinear magnetic structure are quantum materials that exhibit unique spin-dependent properties[7,8], and compounds with chiral magnetic order[9–11] have been intensely investigated due to intriguing topological electronic features. Weyl points close to the Fermi level, serving as sinks or sources of Berry curvature, give rise to an anomalous Hall effect (AHE)[12–14] that enables electrical read-out of the magnetic state. Electrical read-out is also possible via the spin-orbit (SO) coupling induced anisotropic magnetoresistance (AMR), but this is usually small on the order of 0.1%. Large AMR values have been observed at magnetic-field induced transitions between different AFM phases[15] or by current-induced internal fields relying on the strong coupling between charge current and magnetic order[16].

[1]Institute of High Energy Physics, Chinese Academy of Sciences, Beijing 100049, China. [2]Physikalisches Institut, Karlsruhe Institute of Technology, Karlsruhe 76049, Germany. [3]Spallation Neutron Source Science Center, Dongguan 523803, China. [4]Institut für Physik, Johannes Gutenberg Universität Mainz, 55128 Mainz, Germany. [5]Beijing National Laboratory for Condensed Matter Physics, Institute of Physics, Chinese Academy of Sciences, Beijing 100190, China. [6]Songshan Lake Materials Laboratory, Dongguan 523808, China. [7]School of Integrated Circuit Science and Engineering, Beihang University, Beijing 100191, China. [8]NIST Center for Neutron Research, National Institute of Standards and Technology, Gaithersburg, MD 20899, USA. [9]Department of Physics and Beijing Key Laboratory of Opto-electronic Functional Materials & Micro-Nano Devices, Renmin University of China, Beijing 100872, China. [10]Institute for Quantum Materials and Technologies, Karlsruhe Institute of Technology, Karlsruhe 76021, Germany. ✉e-mail: dengsh@ihep.ac.cn; lhhe@iphy.ac.cn; christoph.suergers@kit.edu

Theoretical calculations of the electronic transport properties are most often carried out for single-crystalline materials, where the various anisotropies have to be taken into account. From a materials science point of view, polycrystalline materials are just as interesting as single crystals because they are more widely used in applications, they are compatible with Si-based electronics[8], and crystallographic dependencies are balanced out by the randomly arranged crystallites. In this context, the potential of polycrystalline metals for AFM spintronics has been demonstrated by successful spin-orbit torque switching of polycrystalline AFM heterostructures[17]. Moreover, in the case of a polycrystalline non-collinear antiferromagnet, it is not a priori clear that all non-diagonal elements of the resistivity tensor cancel when subjected to an applied magnetic field, which can change the AFM structure in each crystallite.

Prime examples of compounds where the non-collinear AFM order is susceptible to small variations of composition, strain, and magnetic field are Mn-based $Mn_3AX$ (A=Ga, Ge, Zn, Ag, Ni; X=C, N) compounds of cubic antiperovskite structure. They are well known for their large magnetovolume, piezomagnetic, and barocaloric effects and negative thermal expansion due to a strong spin-lattice

coupling[18–24]. The mutual relation between the crystal lattice and magnetic order is based on the sensitivity of the Mn magnetic moment to the local atomic environment[25]. This leads to a geometrically frustrated lattice with Mn atoms arranged into a kagome lattice in the (111) plane, forming triangles with magnetic moments on equivalent Mn sites[11,26] The long-range AFM order is often of the coplanar $\Gamma^{5g}$ (Fig. 1a, b) or $\Gamma^{4g}$ type[27]. In the latter case, an octupole driven AHE response occurs when the magnetic field is rotated in the kagome plane along related crystallographic directions, which is fundamentally different to the conventional dipole driven AHE[28]. These materials often show rich phase diagrams, and transitions between the phases can be controlled by the magnetic field and current[29].

Here we report an investigation on A-site doped $Mn_3(Zn_{0.5}Ge_{0.5})N$ powder samples where spontaneous changes occur in the longitudinal and transverse resistivity at low temperatures which are susceptible to weak magnetic fields and are even-order (symmetric) functions of the applied magnetic field. We attribute the resistance changes to a magnetic phase transition between different non-collinear phases. The subtle change in magnetic structure is supported by neutron diffraction demonstrating its high sensitivity to minute changes in magnetic

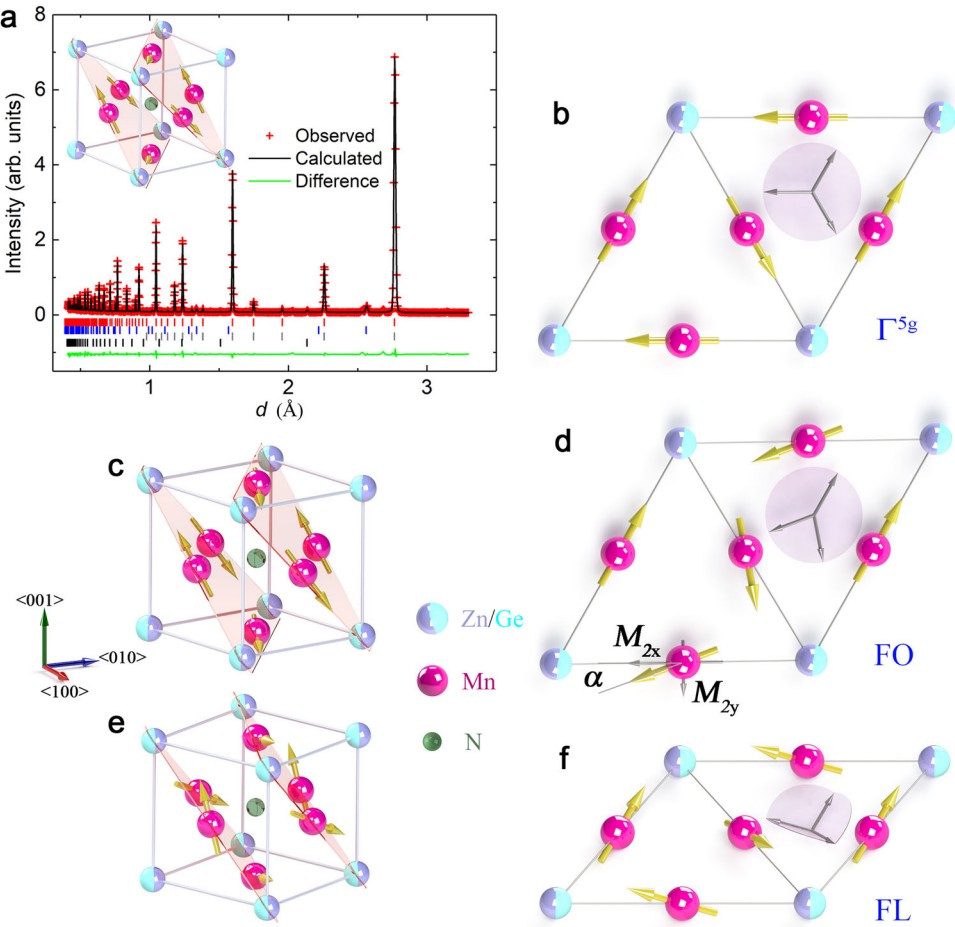

**Fig. 1 | Crystal and magnetic structures of $Mn_3Zn_{0.5}Ge_{0.5}N$. a** Powder neutron-diffraction pattern at $T = 10$ K for the sample of nominal composition $Mn_3Zn_{0.5}Ge_{0.5}N$. The refined composition was determined to be $Mn_3Zn_{0.54}Ge_{0.46}N$ with a lattice constant of 3.9116 Å, see Supplementary Note 3 for details. The vertical markers below the data indicate the angular positions of the nuclear reflections (red, top row), reflections attributed to 3.7 wt% MnO impurities (blue, second row), and magnetic Bragg reflections (black, third row). Contributions attributed to vanadium (green, bottom row) are due to the V cylinder containing the samples during the measurement. Inset displays the antiferromagnetic $\Gamma^{5g}$ spin configuration of cubic $Mn_3Zn_{0.5}Ge_{0.5}N$ antiperovskite. **b** (111) plane showing the non-

collinear coplanar antiferromagnetic "triangle" spin configuration $\Gamma^{5g}$ with zero magnetization. **c, d** The "fork" spin configuration FO with three coplanar magnetic vectors within (111) plane. The angle $\alpha$ indicates the deviation of magnetic moments $M_{2x}$ towards $M_{2y}$ in the (111) plane. **e, f** The "flower" spin configuration FL with three non-coplanar magnetic vectors equally tilted in a direction perpendicular to the (111) plane generating a nonzero magnetization. The arrow length corresponds to the refined value of 3.26 $\mu_B$/Mn. All angles have been exaggerated in the image for clarity. Gray arrows indicate the angular relation of the spins in insets (**b, d, f**). Source data are provided as a Source Data file.

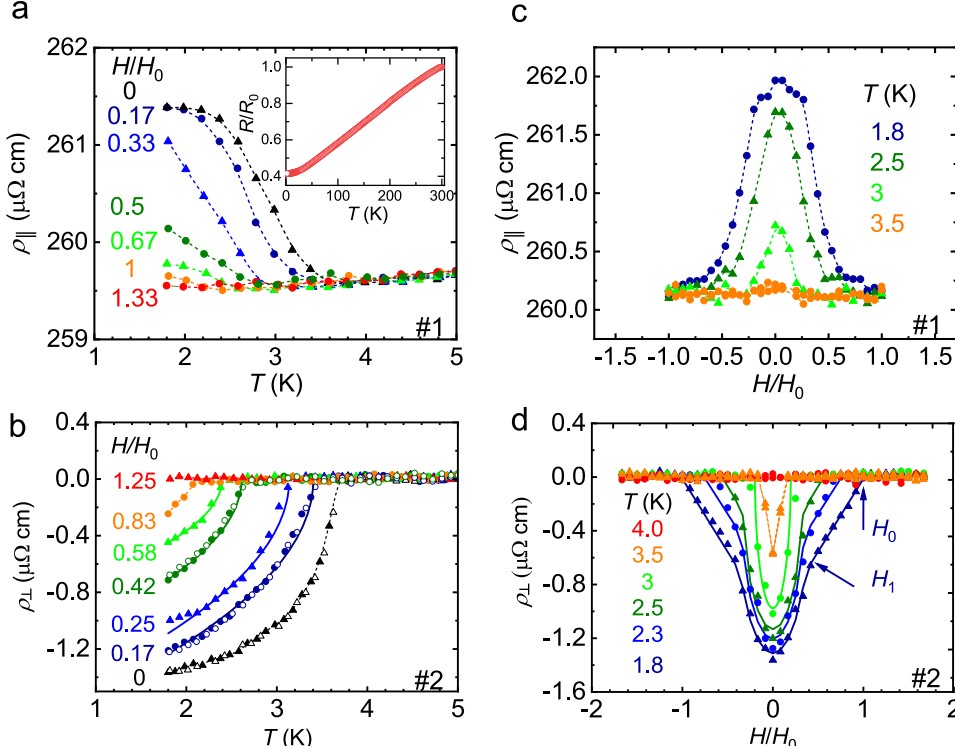

**Fig. 2 | Longitudinal and transverse resistivity. a**, **b** Temperature dependence of the longitudinal resistivity $\rho_\parallel(T)$ and the transverse resistivity $\rho_\perp(T)$ for various magnetic fields $H$ applied perpendicularly to the current and voltage directions for two samples cut from the same batch. Inset shows the $\rho_\parallel(T)$ up to room temperature. **c**, **d** $\rho_\parallel(H)$ and $\rho_\perp(H)$ in perpendicular magnetic field $H$ at various temperatures $T$. Closed (open) symbols indicate data obtained while cooling down (warming up) the sample. The transition field $H_0$ was determined from the resistivity at 1.8 K to 30 mT for **a**, **c** and 120 mT for **b**, **d**, respectively. Broken lines serve as guide to the eye. Solid lines in **b** and **d** represent calculations according to the model, see text. For $\rho_\perp(T)$ and $\rho_\perp(H)$ of sample #1 see Supplementary Note 2. Source data are provided as a Source Data file.

structure. The fact that the magnetic phase transitions between different spin phases are observed in a powder sample, where effects from crystalline anisotropy are cancelled suggests that they are controlled only by exchange interactions, in contrast to the SO interactions controlling, e.g., the $\Gamma^{5g}$ - $\Gamma^{4g}$ transition. This can be adequately described in a phenomenological model revealing the presence of a magnetic-field induced topological orbital momentum. The unexpected and surprising result that in a polycrystal of non-collinear AFM order a finite transverse resistivity remains in zero magnetic field and vanishes in magnetic field is not due to symmetry effects or the AHE and provides information about the nontrivial magnetic phase diagram when properly investigated.

## Results

### Experimental results

The $Mn_3Zn_{0.5}Ge_{0.5}N$ sintered-powder samples with a crystallite size of ~400 nm exhibit the coplanar $\Gamma^{5g}$ type antiferromagnetic order below the Néel temperature $T_N = 411$ K derived from neutron diffraction and magnetization measurements[23,30] (Fig. 1a, Supplementary Fig. 1a and Supplementary Tables 1–3). Refinement of the neutron scattering data at $T = 10$ K confirms the same space group $Pm$-$3m$ and magnetic space group $R$-$3m$ (166,97)[22] as isostructural $Mn_3AN$ (A = Ni, Cu, Zn, Ga, Ge, Pd, In, Sn, Ir, Pt)[25] independent of the doping. Figure 2a and b show the temperature dependence of the longitudinal and transverse resistivities $\rho_\parallel$ and $\rho_\perp$, respectively, of two samples at various magnetic fields $H$. During cooling from room temperature, the sample shows a metallic behavior (inset of Fig. 2a). In zero field, $\rho_\parallel(T)$ and $\rho_\perp(T)$ clearly change upon cooling across a temperature $T^* = 3.7$ K. The non-zero transverse resistivity below $T^*$ even in the absence of a magnetic field is surprising because in a

powder sample with an equiprobable distribution of grain orientations, this component should average out to zero. The step-like transition shifts to lower temperatures and broadens with increasing magnetic field of the order of tens of mT (#1: $H_0 = 30$ mT, #2: $H_0 = 120$ mT). This extraordinary behaviour is unexpected given the usual robustness of AFM order against moderate magnetic fields. In $Mn_3Zn_{0.5}Ga_{0.5}N$ it occurs only close to the phase transition between different non-collinear spin textures, where the Heisenberg exchange is small and the structure is stabilized by the higher order terms of the exchange nature (e.g. biquadratic exchange). Several samples cut from the same batch exhibit an overall similar behavior regarding the step-like behaviour of $\rho_\parallel$ or $\rho_\perp$ at $T^*$. This transition is not observed for isostructural $Mn_3Ag_{0.93}N$ (Supplementary Fig. 5), i.e., in undoped samples and seems to originate from the doping or from the chemical disorder of the A atoms. The corresponding transverse conductivity is $\sigma_\perp(0) = -\rho_\perp/(\rho_\parallel^2 + \rho_\perp^2) = 20$ $\Omega^{-1}$cm$^{-1}$ at $T = 1.8$ K in zero field. This is smaller than the transverse conductivities of 150–500 $\Omega^{-1}$ cm$^{-1}$ arising from an AHE observed for other non-collinear antiferromagnets with triangular magnetic order like single-crystalline $Mn_3Ge$ and $Mn_3Sn$[9,31] but similar to polycrystalline $Mn_3Sn$ films[32,33].

The magnetic field dependences of $\rho_\parallel$ and $\rho_\perp$ below $T^*$ are even functions of the magnetic field (Fig. 2c, d) with a saturation above $H_0$. $\rho_\perp(H)$ is quite different from the magnetic field behavior expected for an AHE which would appear as an odd function of magnetic field due to the broken time-reversal symmetry. We therefore consider $\rho_\parallel(H)$ and $\rho_\perp(H)$ as arising from an AMR = $[\rho(H)-\rho(0)]/\rho_\parallel(0)$-1% which is known to occur in ferromagnetic as well as in AFM materials.[34–36] By investigating several samples cut from the same batch in different directions we observe either an increase or decrease of the

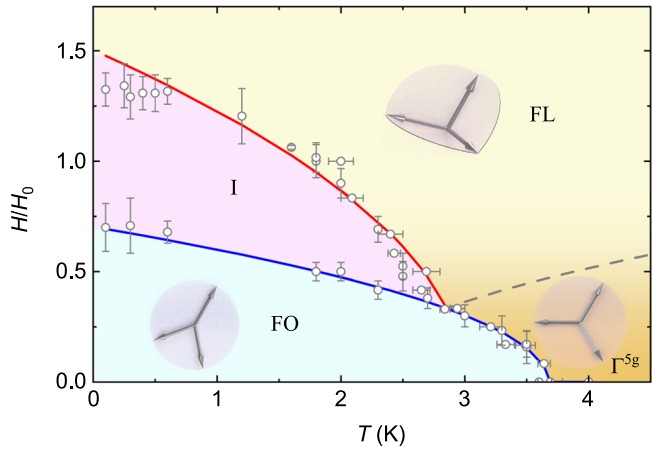

**Fig. 3 | H-T phase diagram derived from the transition in $\rho_\parallel$(T, H) and $\rho_\perp$(T, H) measured down to 0.1 K (see Supplementary Fig. 3).** The boundaries $H_0$ and $H_1$ between the different spin configurations are determined from the change of slope in resistivity as indicated for $T = 1.8$ K in Fig. 2d, specifying the regions of spin configurations "triangle" $\Gamma^{5g}$, "fork" FO, "intermediate" I, and "flower" FL. Solid lines represent the theoretical predictions of the phase diagram. The FL phase is above the highest transition line. The I phase is a mixture of two low-symmetry phases that are obtained due to different components of the magnetic field (in-plane and out-of-plane). Dashed line represents the transition between $\Gamma^{5g}$ and FL which is not observed in the resistivities. Error bars represent the accuracy by which the data could be determined from the $\rho$ (T, H) plots. Source data are provided as a Source Data file.

resistivities while crossing $T^*$. We could not check whether this behaviour could be changed by, e.g., magnetic-field cooling of the sample from above $T_N$[34,35] because the sample degraded at higher temperatures close to $T_N$ and the effect disappeared. Apart from the different signs, both longitudinal and transverse resistivity components exhibit the same temperature and field dependence.

The size of $\rho_\perp$ is independent of the magnetic field orientation (Supplementary Fig. 4) when the field is rotated from out of plane to in plane (angle $\varphi$) or in the plane around the surface normal (angle $\omega$) of the sample. A sinusoidal dependence of the AMR amplitude on the angle between current and magnetic field is not observed possibly due to the polycrystallinity of the sample strongly reducing the orientation dependence of the AFM[1].

The observed transition cannot be ascribed to superconductivity[37] or to a structural phase transition, which is confirmed by the nonzero $\rho_\parallel$ and neutron scattering. The hysteresis of the resistivity when cooling and heating the sample (Fig. 2b) or sweeping the magnetic field up and down (Supplementary Fig. 3b) is almost zero or negligibly small, respectively. Effects arising from weak localization or weak antilocalization in a 3D metal or even in a 3D Weyl metal do not play a role[38,39] because the expected corrections to the conductivity in dependence of $T$ and $H$ are not observed. The step-like behavior of $\rho_\parallel$(T,H) and $\rho_\perp$(T,H) and its saturation at moderate magnetic fields indicate a transition of magnetic origin rather than weak localization effects.

Measurements of the magnetoresistance down to 0.1 K establish a H-T phase diagram with phase boundaries appearing between different spin configurations (Fig. 3 and Supplementary Fig. 3c) discussed in detail below. Interestingly, the transverse component of resistivity is zero at high magnetic fields above a critical value, as well as in the fully compensated antiferromagnetic phase $\Gamma^{5g}$. This observation seems contradictory, but our analysis of the temperature- and field-dependencies of resistivity points to the existence of two phase-transition lines (Fig. 3, red and blue lines) instead of one.

The integral magnetization $M$ of $Mn_3Zn_{0.5}Ge_{0.5}N$ is very small and does not show a transition at low temperatures at 2 K apart from a shallow increase below 5 K (Supplementary Fig. 1b). From the temperature dependence of $M$ we estimate a tiny average magnetic moment of $3.6 \times 10^{-5}$ $\mu_B$/Mn at $T = 2$ K in a weak magnetic field of 10 mT and $0.4 \times 10^{-5}$ $\mu_B$/Mn from the magnetization $M(0)$ in zero field (Supplementary Fig. 1).

In addition, neutron diffraction was employed to confirm the subtle change of the magnetic phase with temperature. The positions of the reflections in the diffraction patterns of $Mn_3Zn_{0.5}Ge_{0.5}N$ are independent of temperature between 2 K and 10 K (Supplementary Fig. 6), implying that variations of the crystal and/or magnetic structures with temperature are very small. However, the refinement of the diffraction peaks reveals the magnitude and direction of the magnetic moments in the long-range magnetically ordered state. In the present case, an intensity difference $I(T) - I(10$ K$)$ of a few percent is attributed to a change of magnetic texture with decreasing temperature (Fig. 4a). In particular, the considerable increase of intensities at $P_3$ and $P_4$ below 4 K is correlated with the appearance of the additional contribution to the resistivities below $T^* = 3.7$ K. We estimate the relative contributions of nuclear and magnetic scattering by Rietveld analysis, revealing no magnetic contributions for $P_1$, $P_2$ but approximately 5% and 11% for the intensities of $P_3$, and $P_4$, respectively, at 2 K (Fig. 4b, inset). In Fig. 4b, the integrated intensities at $P_3$ and $P_4$ - including a magnetic contribution – first only gradually increase with cooling but then strongly increase below 5 K while intensities at $P_1$ and $P_2$ remain almost zero. This indicates that the local magnetic moments of Mn atoms slightly change as the temperature drops below 3.7 K without a change of the crystalline lattice. The neutron diffraction pattern at 10 K (Supplementary Table 2) can be fitted well with a structural model of cubic symmetry (Fig. 1a) and a superlattice magnetic model of $\Gamma^{5g}$ type displayed in Fig. 1b, resulting in a refined moment $M_{2x} = 3.26(2)$ $\mu_B$/Mn. However, if we use the same model to fit the patterns below 10 K, the refinement becomes worse. Assuming a finite magnetic moment component $M_{2y}$ along the $y$ direction (Fig. 1d) clearly improves the profile difference. $M_{2y}$ is determined from the minimalization of the refinement parameter $\chi^2$ (Supplementary Fig. 8). The evolution of spin texture with cooling below 3.7 K is shown in Fig. 4c. $M_{2y}$ increases towards low temperatures while $M_{2x}$ remains almost constant. The moment reorientation from the perfect triangle configuration ($\Gamma^{5g}$, $\alpha = 0$) towards the new configuration ($\alpha > 0$) exemplified by the angle $\alpha$, gradually grows from zero at 4 K to 6.5° at 2 K (Fig. 4 c) and the net magnetic moment increases by 1%. Hence, the coplanar triangle $\Gamma^{5g}$ phase (Fig. 1b) changes to a coplanar magnetic phase FO (Fig. 1d) of lower symmetry.

The strong link between the variation of the resistivities $\rho$ and the variation of neutron diffraction intensities $P_3$ and $P_4$ across the phase transition is corroborated by comparison with the results for isostructural $Mn_3AgN$, where neither a variation of $\rho$ nor a variation of intensities $P_3$ and $P_4$ with temperature are observed (Supplementary Figs. 5, 10). For $Mn_3AgN$, we do not observe a variation of the intensities $P_i$ ($i = 1$–4) upon cooling to 2 K within the accuracy of the measurement in agreement with the lack of a transition in the resistivity (Supplementary Figs. 5, 10, 11). In this case, the triangle configuration remains stable.

Application of a magnetic field at $T = 3$ K $< T^*$, gives rise to a broad dip around 20 mT in the intensity difference $I(\mu_0 H) - I(20$ mT$)$ of the integrated intensity $P_4$ while $P_2$ remains constant (Fig. 4d, Supplementary Fig. 9). The decrease of the integrated intensity $P_4$ in from zero to 20 mT is attributed to a field induced change of the magnetic phase in accordance with the phase diagram in Fig. 3. These results confirm the change of the coplanar triangular spin arrangement with decreasing temperature and/or increasing field and demonstrate the high sensitivity of the neutron powder diffraction to small changes in the magnetic structure.

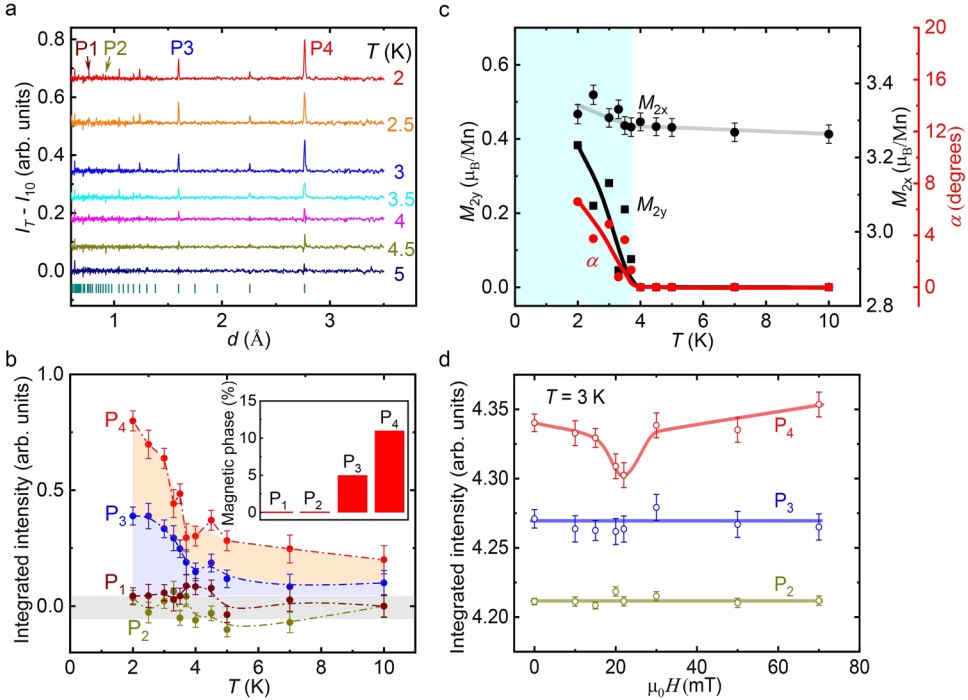

**Fig. 4 | Neutron diffraction patterns of Mn$_3$Zn$_{0.5}$Ge$_{0.5}$N. a** Intensity difference $I_T–I_{10} = I(T)\cdot I(10\,K)$ of neutron diffraction patterns. The data are shifted with respect to each other for clarity. **b** Temperature dependence of the integrated intensity of reflections P$_1$, P$_2$, P$_3$, and P$_4$, occurring at 0.78 Å, 0.93 Å, 1.60 Å, and 2.77 Å, respectively, in **a**. For clarity, data of P$_3$ and P$_4$ are shifted upward by 0.1 and 0.2, with respect to P$_1$ and P$_2$, respectively. The amount of magnetic phase estimated from the Rietveld refinement is displayed in the inset. **c** Temperature dependence of the local magnetic moments $M_{2x}$ and $M_{2y}$ of Mn atoms and the angle $\alpha$ determined by Rietveld refinement where a nonzero $M_{2y}$ indicates the direction of a deviation from the triangle $\Gamma^{5g}$-type order. The temperature range where the spontaneous magnetoresistivity occurs is indicated by a cyan colored background. The lines in the figure are guides to the eye. **d** Field dependence of the integrated intensity of reflections P$_2$, P$_3$, and P$_4$ at $T = 3\,K < T^*$. Intensities of P$_2$ and P$_3$ have been shifted upwards by 3.95 and 1.07 respectively, for clarity. Error bars represent the standard deviation. Source data are provided as a Source Data file.

## Theoretical model

The experimentally observed field-induced changes in the diagonal and off-diagonal elements of the resistivity tensor are in stark contrast to the analogous dependence for ferromagnets. In ferromagnetic polycrystalline or powder samples, the magnetic field induces a net magnetization pointing in the same direction. The transverse (with respect to the current) resistivity component is odd in the magnetization and as such: (i) vanishes in the absence of the field due to averaging over the grains; (ii) depends on the orientations of the magnetic field; (iii) increases with increasing magnetic field values. In contrast, in Mn$_3$Zn$_{0.5}$Ga$_{0.5}$N we observe that (i) $\rho_\perp$ appears below the critical temperature of 3.7 K even in the absence of the magnetic field; (ii) $\rho_\perp$ is independent of the field orientation; (iii) $\rho_\perp$ is non-zero in a finite range of magnetic field values and vanishes above some critical field. This difference leads to the hypothesis that Mn$_3$Zn$_{0.5}$Ga$_{0.5}$N undergoes the phase transition between three magnetic phases with different spin symmetries, i.e., with different relative angles between the neighboring spins[40,41]. In addition, we believe that the magnetic field induces interacting topological orbital momenta[42] whose contribution to the sample energy dominates the contribution of the field-induced magnetization. Our results suggest that the energy associated with the topological orbital momentum gives rise to an effective ponderomotive force. This force is proportional to the square of the magnetic field and serves to stabilize the highly symmetric phases, even in the presence of the magnetic field as discussed below.

In the following, we develop a phenomenological model to describe the observed field and temperature dependencies of the resistivity tensor. The model is based on analyzing the spin symmetries of the existing phases, as illustrated in Fig. 1b, d, and f. We distinguish between two high-symmetry phases: coplanar ('triangle' $\Gamma^{5g}$, Fig. 1b) and noncoplanar ('flower' FL, Fig. 1f), as well as one coplanar low-symmetry phase ('fork' FO, Fig. 1d). Formally, these phases can be described in terms of three spin (or magnetization) vectors $\mathbf{M}_{1,2,3}$ localised at different magnetic atoms within unit cell. From symmetry point of view it is convenient to introduce linear combinations $\mathbf{N}_1 = (\mathbf{M}_1 + \mathbf{M}_2 - 2\mathbf{M}_3)/\sqrt{6}$ and $\mathbf{N}_2 = (-\mathbf{M}_1 + \mathbf{M}_2)/\sqrt{2}$, that form a multidimensional order parameter of the antiferromagnetic phases, and the combination $\mathbf{M} = (\mathbf{M}_1 + \mathbf{M}_2 + \mathbf{M}_3)/\sqrt{3}$, which is the total magnetization[43,44]. The properties of the phases are summarized in Table 1.

Our model is based on the analysis of the magnetic energy $F$ which, in the spirit of the Landau's theory of phase transitions, we introduce in a following way (for details see Supplementary Note 5)

$$F = \frac{1}{2}J(T)\mathbf{M}^2 + \frac{1}{4}D\mathbf{M}^4 - \frac{1}{2}D'\left[(\mathbf{N}_1 \cdot \mathbf{M})^2 + (\mathbf{N}_2 \cdot \mathbf{M})^2\right] \\ - \Lambda_{CC}\left[(\mathbf{N}_1 \cdot \mathbf{H})^2 + (\mathbf{N}_2 \cdot \mathbf{H})^2\right] - \mathbf{MH} \tag{1}$$

The first three terms of the equation represent the bilinear (Heisenberg) and the biquadratic exchange coupling. The strength of coupling is characterized by phenomenological constants $J(T)$, $D$, and $D'$. We assume a linear temperature dependence of $J(T) = \beta T$ with $\beta > 0$.

The next to last term in Eq. (1) requires special discussion. We attribute this effect to the energy of chiral-chiral interactions, which create correlations between the emergent topological orbital momenta in non-coplanar magnetic structures. According to symmetry analysis, these momenta can appear in non-coplanar Mn$_3$Zn$_{0.5}$Ge$_{0.5}$N due to field or temperature induced fluctuations in

**Table 1 | Properties of the spin phases**

| Phase | Spin Symmetry | Ordering | Magnetic vectors | Resistivity tensor | Figure |
|---|---|---|---|---|---|
| Triangle ($\Gamma^{5g}$) | $D_{3d}$ | Coplanar | $\mathbf{N_1} \perp \mathbf{N_2}$, $\|\mathbf{N_1}\| = \|\mathbf{N_2}\|$, $\mathbf{M} = 0$ | $\rho_{xx} = \rho_{yy} \neq \rho_{zz}$ | 1b |
| Flower | $C_3$ | Noncoplanar | $\mathbf{N_1} \perp \mathbf{N_2}$, $\|\mathbf{N_1}\| = \|\mathbf{N_2}\|$, $\mathbf{M} \perp \mathbf{N_1}, \mathbf{N_2}$ | $\rho_{xx} = \rho_{yy} \neq \rho_{zz}$ | 1f |
| Fork | $C_2$ | Coplanar | $\mathbf{M} \| \mathbf{N_1} \perp \mathbf{N_2}$, $\|\mathbf{N_1}\| \neq \|\mathbf{N_2}\|$ | $\rho_{xx} \neq \rho_{yy} \neq \rho_{zz}$ | 1d |

Spin symmetry group operations and components of the resistivity tensor $\rho_{ij}$ are defined in the spin-related frame: the $z$ axis is perpendicular to the ordering plane spanned by the vectors $\mathbf{N_1}$ and $\mathbf{N_2}$ (or is along 3rd order axis), the $x$ axis is parallel to the vector $\mathbf{N_1}$ (or is along 2nd order axis). For the resistivity tensor only non-zero components are given.

the relative orientation of the spins. As reported in ref. 42, the energy of the chiral-chiral interactions can be expressed as $E_{CC} = \kappa_{CC}(\mathbf{M} \cdot \mathbf{N_1} \times \mathbf{N_2})^2$. From the observed coplanar ordering in the absence of a magnetic field, we conclude that the effective chiral-chiral coupling strength $\kappa_{CC} > 0$ in Mn$_3$Zn$_{0.5}$Ge$_{0.5}$N is positive. However, in the presence of an external magnetic field, the energy is modified to $E_{CC} = \kappa_{CC}\chi^2(\mathbf{H} \cdot \mathbf{N_1} \times \mathbf{N_2})^2$, where $\chi$ represents the magnetic susceptibility. Further transformation of this expression can be achieved using relations between the magnetic vectors $\mathbf{M}, \mathbf{N_1}, \mathbf{N_2}$ (see Supplementary Note 5 for details), leading to the final term in the equation.

To obtain magnetic configurations corresponding to the triangle, flower, and fork phases, we minimized the energy Eq. (1) for a given field and temperature. Based on our analysis, we find that at zero field, the high temperature $\Gamma^{5g}$ phase is unstable and transforms to the low-symmetry FO phase at $T^* = T_c = 3D'/(2\beta M_s^2)$, where $M_s$ is sublattice magnetization. This conclusion is supported by the temperature dependence of the resistivity, as shown in Fig. 2a, b. If an external magnetic field is applied parallel to the ordering plane, it creates a ponderomotive force that adds an additional pressure, which stabilizes the FO phase within a temperature range $T < T_c - \Lambda_{CC}H^2/(3\beta M_s^2)$. However, in the same magnetic field configuration, we observe that the $\Gamma^{5g}$ phase becomes unstable and transforms to the non-coplanar FL phase at $T \geq \Lambda_{CC}H^2/(3\beta M_s^2)$. Hence, at the triple point $H_{cr} \equiv M_s\sqrt{3\beta T_c/\Lambda_{CC}}$, the transition to the FO phase occurs simultaneously with the growth of the noncoplanar magnetization component. In this case, the transition line is $T < T_c - \tilde{\Lambda}_{CC}H^2/(3\beta M_s^2)$, where $\tilde{\Lambda}_{CC} < \Lambda_{CC}$ (line separating $\Gamma^{5g}$ and intermediate phase $I$ in Fig. 3).

The above considerations are relevant for interpreting the phase diagram of a single crystalline sample. However, to analyze the resistivity data and extract the phase diagram of a powder sample consisting of single-domain particles, we assume an equiprobable distribution of particle orientations. In addition, we employ symmetry considerations, assuming that $\rho_{xx} - \rho_{yy} \propto (\mathbf{N_I M})^2$ determines the field and temperature dependencies of the AMR. We also neglect any variations in the component $\rho_{zz}$ and consider its value to be the same in all three phases. We compute the resistivity tensor for each phase by considering all possible orientations of the magnetic field with respect to crystallographic axes. After computing the average over the particle distribution, we obtain a phase diagram that can be divided into three distinct regions (Fig. 3).

## Discussion

Within the first region, denoted as $\Gamma^{5g}$, either the triangle or FL high-symmetry phase remains stable. These two phases have identical resistivity tensors, which makes it impossible to distinguish between them and resolve the transition line (Fig. 3, dashed line) separating them. In the FO phase, all particles underwent a transition from the triangle phase to the FO phase, followed by a smooth growth of in-plane magnetization $M_y$, and the associated field and temperature dependencies of the AMR exhibit a smooth dependence. Conversely, in the intermediate phase $I$, some of the particles transform from the FL phase into the low symmetry phase with decreasing magnetic field. Consequently, the AMR exhibits a kink (Fig. 2d) when the contribution of these particles becomes noticeable in the signal.

In the polycrystalline material, different grains within the sample are affected by different components of the magnetic field. The transition from the fully compensated $\Gamma^{5g}$ phase to the low-symmetry FO phase is induced by the field component that is parallel to the {111} ordering planes. On the other hand, the transition to the high symmetry non-coplanar FL phase is induced mainly by the field component that is perpendicular to the ordering plane, and additionally stabilized by the in-plane component in the vicinity of the triple point $H_{cr}(2.8K)$. Importantly, we find that the transition between the noncoplanar FL and low-symmetry phases occurs at a higher field strength compared to the transition from $\Gamma^{5g}$ to the low-symmetry FO phase. Consequently, the phase diagram of the polycrystalline sample comprises two distinct phase transition lines that delineate the transitions into the low-symmetry phase from $\Gamma^{5g}$ and from the FL phase into the $I$ phase, respectively.

Finally, we discuss the origin of the nonzero transverse resistivity component. The spin symmetry, which describes the effects of exchange interactions, can only account for diagonal components of the resistivity tensor. However, the contribution of magnetic symmetry, which captures the effects of SO interactions, cannot be ignored as it can lead to magnetic order and subsequently affect the resistivity tensor. Therefore, the influence of both spin and magnetic symmetries on the resistivity tensor needs to be considered in order to fully understand the transverse resistivity component. We performed a qualitative analysis of the magnetic symmetry in all three phases. The magnetic symmetry group of the high-symmetry triangle and flower phases still includes third-order rotations, which results in the exclusion of transverse resistivity components in these cases. However, in the low-symmetry FO phase, the appearance of in-plane magnetization is related to the rotation of spins out of the {111} crystallographic planes[43,44]. This symmetry reduction allows for the presence of transverse components of the resistivity. In particular, a nonzero non-diagonal component of the resistivity tensor occurs only in the FO (and $I$) phases (for details of the derivation see Supplementary Note 5) in agreement with the experimental data.

Furthermore, the magnitude and direction of these components depend on the orientation of the magnetic field relative to the sublattice magnetization rather than the crystallographic axes. Therefore, the transverse component of the resistivity is not averaged out in a powder sample (see Supplementary Note 5).

## Methods

### Sample preparation

Sintered polycrystalline samples of the composition Mn$_3$Zn$_{0.5}$Ge$_{0.5}$N and Mn$_3$Ag$_{0.95}$N were prepared by solid-state reaction using fine powders of Mn$_2$N, Zn, Ge, and Ag as starting materials[21]. The well-mixed powders in stoichiometric proportion were pressed into pellets and then sealed in a quartz tube under vacuum ($10^{-5}$ Pa) by wrapping in a Ta foil. The quartz tube was sintered in a box furnace at 1073 K for 80 hours, and then cooled down to room temperature.

### Sample characterization

The crystalline and magnetic structure was characterized by the time-of-flight (TOF) diffractometer GPPD (general purpose powder diffractometer) at CSNS (China Spallation Neutron Source), Dongguan,

China. The neutron beam of GPPD with the high intensity and good spatial resolution makes it well suited for the accurate analysis of magnetic materials with complex orderings[45]. Samples were loaded in a vanadium holder and installed in a liquid helium cryostat that achieves temperatures down to 2 K and magnetic fields up to 70 mT. The neutron diffraction patterns were collected in the TOF mode with wavelength bands of 0.1–4.9 Å between 2 K and 300 K. The General Structure Analysis System (GSAS) program was used for Rietveld refinement with scattering lengths of $-0.373 \times 10^{-12}$, $0.568 \times 10^{-12}$, $0.818 \times 10^{-12}$, $0.592 \times 10^{-12}$, and $0.936 \times 10^{-12}$ cm for Mn, Zn, Ge, Ag, and N, respectively[46]. The compositions determined by the Rietveld analysis of the neutron diffraction data at 300 K are $Mn_3Zn_{0.54}Ge_{0.46}N$ and $Mn_3Ag_{0.93}N$ respectively.

Electronic-transport properties were measured in a physical property measurement system (PPMS, Quantum Design) for temperatures 1.8–400 K and in magnetic fields up to 9 T. Measurements down to 0.1 K have been taken in a $^3$He/$^4$He dilution refrigerator using a Lakeshore 370 ac resistance bridge. The longitudinal resistivity $\rho_{\parallel}$ and the transverse resistivity $\rho_{\perp}$ were measured on samples of rectangular shape with 0.12-mm thick Cu wires glued by silver epoxy. A current $I$ was passed through the sample along the long side of the sample and voltages $V_{\parallel}$ and $V_y$ were measured along and perpendicular to the current direction, respectively, to yield $\rho_{\parallel} = V_{\parallel} wt/I\,l$ and $\rho_{\perp} = V_{\perp}\,t/I$, where $w$, $t$, and $l$ are the width, thickness, and distance between the voltage contacts, respectively. The magnetic field $H$ was applied perpendicularly to both $\parallel$ and $\perp$ directions except for angle-dependent measurements where it was rotated towards the $\parallel$ and $\perp$ directions. Temperature-dependent magnetization data between 2 and 30 K and isothermal magnetization curves were recorded in a superconducting quantum-interference device (SQUID) magnetometer for magnetic fields up to 5 T. In addition, the temperature dependence of magnetization between 300 and 580 K was also measured by a vibration sample magnetometer (VSM).

## Data availability
Source data are provided with this paper. Other data is available from the authors upon request. Source data are provided with this paper.

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

## Acknowledgements

We thank S. Meissner, A. Meissner, and G. Weiß for help with electronic transport measurements below 2 K, and thank H. L. Lu, J. Z. Hao, and S. Srichandan for helpful discussions. We gratefully acknowledge early theoretical calculations by I. Samathrakis and H. Zang. S.D. acknowledges support from Sino-German Postdoc Scholarship Program, National Natural Sciences Foundation of China (NSFC) (52371190), the Guangdong Basic and Applied Basic Research Foundation (2022A1515140117) and the Large Scientific Facility Open Subject of Songshan Lake, Dongguan, Guangdong. J.C. acknowledges funding by the NSFC (U2032220). C.W. acknowledges funding by the NSFC (52272264). R.Z. acknowledges support from the NSFC (11974405). J.S. acknowledges funding by Grant Agency of the Czech Republic grant no. 19-28375X. This work was supported by the Sino-German Mobility Programme No. M-0273 and by the Deutsche Forschungsgemeinschaft (DFG) through CRC TRR 288 – 422213477 "ElastoQMat" (Projects A08 and A09). List of grants: NSFC 52371190, S.D. Guangdong Basic and Applied Basic Research Foundation 2022A1515140117, S.D. Large Scientific Facility Open Subject of Songshan Lake, S.D. NSFC U2032220, J.C. NSFC 52272264, C.W. NSFC 11974405, R.Z. Grant Agency of the Czech Republic 19-28375X, J.S. Sino-German Mobility Programme No. M-0273, S.D., C.W., C.S., G.F. DFG TRR 288 – 422213477 "ElastoQMat" (A08, A09), W.W., C.S., J.S., O.G. We acknowledge support by the KIT-Publication Fund of the Karlsruhe Institute of Technology.

## Author contributions

S.D. and C.S. conceived and designed the experiments. S.D. prepared the samples with the help of C.W., and S.D. carried out the transport measurements. L.H., J.C., F.S., Z.T. and S.D. performed neutron diffraction measurements and the related data analysis with the help of Q.H. G.F. measured the magnetization. R.Z. and Z.H. conducted the NMR experiments. O.G. developed the theoretical model. O.G., L.Š., J.S., W.W., S.D. and C.S. analyzed the data and wrote the manuscript with contributions from all authors.

## Funding

## Competing interests

The authors declare no competing interests.
