## [Peer Review File · Nature Communications]

Deng et al. demonstrated transition of magnetic phases of different symmetry in polycrystal $\text{Mn}_3\text{Zn}_{0.5}\text{Ge}_{0.5}\text{N}$ and associated the transition with topological orbital momenta. It is novel that the polycrystalline samples show phase transition controlled by the exchange interaction under the magnetic field. I might recommend the publication if these points are clarified:

1. More interpretation and references should be added to support these arguments.
 - (a) The appearance and disappearance of these components (additional components of the resistivity tensor) indicates a transition between low- and high-symmetry phases. (Line 209, 210)
 - (b) In addition, we employ symmetry considerations, assuming that $\rho_{xx} - \rho_{yy} \propto (N_1 \cdot M)^2$ determines the field and temperature dependencies of the AMR. (Line 264, 265)
2. The author associated the transition of magnetic phases with the topological orbital momenta (Line 236 - 238). More evidence than the similar energy form should be given to support the association between the magnetic phase transition and the topological orbital momenta. Ref. 37 only proposed the topological-chiral interaction in FeGe and MnGe. Why does $\text{Mn}_3\text{Zn}_{0.5}\text{Ge}_{0.5}\text{N}$ have the similar effects and features?
3. Does the neutron diffraction reflect the moments in each sublattice or the antiferromagnetic order parameter (Neel vector or the magnetic octupole)? The authors mentioned that effects from crystalline anisotropy are cancelled while effects from exchange interaction remain. Why are the effects from magnetic structures not cancelled in polycrystals? Are the effects under the magnetic field just from the remanence instead of different magnetic structures? How to distinguish the effects of the magnetic structure from the remanence?
4. The theory model analyze the effects at a macroscopic level, which is usually used for single crystal and improper for polycrystals. The arguments for polycrystals is too casual in Line 260-270. More calculation details should be given, and simulation for polycrystal and multi-domains should be performed.
5. The M-H curve should be added to support the average magnetic moments (Line 159). Does the curve have a diamagnetic or paramagnetic background? Is the background deducted when calculating the average magnetic moments? What is the magnetic field the authors used to calculate the average magnetic moment of Mn atoms?
6. Are the R-T curves measured during cooling process or heating process? Curves from both processes should be given to exclude the transition of the structure as a double check.

Reviewer #2 (Remarks to the Author):

Deng et al. reported a study on magnetic phase transitions in a non-collinear antiferromagnet (NCAFM) $\text{Mn}_3\text{Zn}_{0.5}\text{Ge}_{0.5}\text{N}$. The main method is longitudinal and transverse resistivity measurements under magnetic fields, where the boundaries of the R-H curve indicate the phase transition points. The experimental results are explained by a phenomenological model on the topological orbital momenta induced by magnetic fields.

NCAFM are attracting increasing interest due to their non-trivial spin and electronic structures. Particularly, many NCAFM exhibit multiple magnetic phases, whose magnetic and electronic properties are still unclear, posing challenges to advance the fundamental understanding and implementation of NCAFM spintronics. An in-depth study that determines the phase transition diagram and the characteristic magneto-transport effects, as done in this study, is of certain interest and importance, thus being potentially publishable in Nature Communications. However, the following comments should be addressed before I can recommend for publication.

1. Line 97: It has been reported that polycrystalline NCAFM Mn_3Sn exhibits a large AHE. I don't think the discussion that excludes "symmetry effects and AHE" from the origin of the transverse resistivity is convincing enough. More evidence or analysis should be provided.
2. Fig. 2: The longitudinal and transverse resistivity was taken from two samples. Is there a specific reason for not measuring in the same sample? How does the sample-to-sample variation influence the conclusion?
3. Line 59: The authors attributed the MR effects in refs. 14 and 15 as large or even "colossal" AMR, whereas their MR is on the order of 0.1%. Please reconsider the terminology.
4. Line 116: The transition field on the order of 10 to 100 mT is actually far below the expected robustness of an AFM. What is the reason?
5. Line 137: It is surprising that the transverse resistivity does not depend on the magnetic field orientation, which is different from the polycrystalline FM samples. The authors should explain more explicitly on this independence, rather than simply ascribe it to the polycrystalline structure of the AFM.
6. Following comment 5, does the longitudinal resistivity show any angle dependence?
7. Line 242: When a magnetic field H is applied to the NCAFM, M is replaced by χH , while N_1 and N_2 remain unchanged. Will N_1 and N_2 also be functions of H ?
8. The authors mentioned the uniqueness of using polycrystalline AFM materials for spintronic applications. I suggest the authors to enrich the discussions by referring to the relevant content and references in a recent review Nat. Mater. 22, 684–695 (2023).

Reply to the Reviewers

We would like to thank the Reviewers for the valuable time they invested in preparing their reviews and for their stimulating comments. Below we have answered their questions in a point-by-point response. Our responses and the changes in the main text and in the Supplementary Information (SI) are highlighted in red color. For the sake of simplicity, all changes in the manuscript and in the SI are repeated in italics in this response.

Reviewer #1

Deng et al. demonstrated transition of magnetic phases of different symmetry in polycrystal $\text{Mn}_3\text{Zn}_{0.5}\text{Ge}_{0.5}\text{N}$ and associated the transition with topological orbital momenta. It is novel that the polycrystalline samples show phase transition controlled by the exchange interaction under the magnetic field. I might recommend the publication if these points are clarified:

1. More interpretation and references should be added to support these arguments.

(a) The appearance and disappearance of these components (additional components of the resistivity tensor) indicates a transition between low- and high-symmetry phases. (Line 209, 210).

Answer: We have removed this phrase and have rewritten the first paragraph of the theoretical part (lines 222-238) to address this issue and the next comments:

The experimentally observed field-induced changes in the diagonal and off-diagonal elements of the resistivity tensor are in stark contrast to the analogous dependence for ferromagnets. In ferromagnetic polycrystalline or powder samples, the magnetic field induces a net magnetization pointing in the same direction. The transverse (with respect to the current) resistivity component is odd in the magnetization and as such: (i) vanishes in the absence of the field due to averaging over the grains; (ii) depends on the orientations of the magnetic field; (iii) increases with increasing magnetic field values. In contrast, in $\text{Mn}_3\text{Zn}_{0.5}\text{Ga}_{0.5}\text{N}$ we observe that (i) ρ_{\perp} appears below the critical temperature of 3.7 K even in the absence of the magnetic field; (ii) ρ_{\perp} is independent of the field orientation; (iii) ρ_{\perp} is non-zero in a finite range of magnetic field values and vanishes above some critical field. This difference leads to the hypothesis that $\text{Mn}_3\text{Zn}_{0.5}\text{Ga}_{0.5}\text{N}$ undergoes the phase transition between three magnetic phases with different spin symmetries, i.e., with different relative angles between the neighboring spin^{40,41}. In addition, we believe that the magnetic field induces interacting topological orbital momenta⁴² whose contribution to the sample energy dominates the contribution of the field-induced magnetization.

(b) In addition, we employ symmetry considerations, assuming that $\rho_{xx} - \rho_{yy} \propto (\mathbf{N} \cdot \mathbf{M})^2$ determines the field and temperature dependencies of the AMR. (Line 264, 265)

Answer: We are grateful to the Reviewer for this comment on the transverse resistivity. To clarify our arguments, we have added the derivation of the transverse component for a single crystal and a powder sample to the Supplementary Information Note 5. We have also added details of the averaging procedure and modified the main text (lines 334-344) correspondingly:

However, in the low-symmetry FO phase, the appearance of in-plane magnetization is related to the rotation of spins out of the $\{111\}$ crystallographic planes^{43,44}. This symmetry reduction allows for the presence of transverse components of the resistivity. In particular, a nonzero nondiagonal component of the resistivity tensor occurs only in the FO (and I) phases (for details of the derivation see Supplementary Information, Note 5) in agreement with the experimental data.

Furthermore, the magnitude and direction of these components depend on the orientation of the magnetic field relative to the sublattice magnetization rather than the crystallographic axes. Therefore, the transverse component of the resistivity is not averaged out in a powder sample (see Supplementary Information, Note 5).

2. The author associated the transition of magnetic phases with the topological orbital momenta (Line 236 - 238). More evidence than the similar energy form should be given to support the association between the magnetic phase transition and the topological orbital momenta. Ref. 37 only proposed the topological-chiral interaction in FeGe and MnGe. Why does $\text{Mn}_3\text{Zn}_{0.5}\text{Ge}_{0.5}\text{N}$ have the similar effects and features?

Answer: In this paper we develop a phenomenological model in the spirit of the Landau theory of phase transitions. This model is based on the symmetry analysis of the different magnetic textures and thus allows a rather general, symmetry-based classification of different interactions (e.g. exchange vs spin-orbit). The same approach can be extended to the study of topological orbital momenta, the existence of which can be predicted from symmetry considerations for noncollinear antiferromagnetic structures. This, by the way, is how it was done in Ref. [37], where the topological orbital momenta first appeared as a result of the symmetry analysis and then their existence was proved numerically for the particular materials. In this paper we propose the mechanism related to topological orbital momenta as a plausible explanation, although it can be not the only one.

To address these issues, we modified the text in lines 222-238 (see our answer to your comment 1) and, in addition, in lines 262-267 as follows:

We attribute this effect to the energy of chiral-chiral interactions, which create correlations between the emergent topological orbital momenta in non-coplanar magnetic structures. According to symmetry analysis, these momenta can appear in a non-coplanar $\text{Mn}_3\text{Zn}_{0.5}\text{Ge}_{0.5}\text{N}$ due to field or temperature induced fluctuations in the relative orientation of the spins. As reported in Ref. 42, the energy of the chiral-chiral interactions can be expressed as...

3. Does the neutron diffraction reflect the moments in each sublattice or the antiferromagnetic order parameter (Neel vector or the magnetic octupole)?

Answer: Neutron diffraction can be used to directly measure the atomic arrangement of magnetic moments in a bulk antiferromagnetic crystal. In powder diffraction, the diffraction intensity I_{hkl} is proportional to $|F_{hkl}|^2$ where F_{hkl} is the amplitude of the diffracted X-ray or neutron hkl reflection.

For X-ray diffraction

$$F_{hkl} = \sum_j f_j \cdot \exp(2\pi i(hx + ky + lz)) \cdot e^{-2W} \quad (1)$$

where f_j is the X-ray atomic scattering factor of atom j for X-rays and W is a thermal correction to F_{hkl} .

For magnetic neutron scattering,

$$F_{hkl} = \sum_j q_j \cdot f_{Mj} \cdot \exp(2\pi i(hx + ky + lz)) \cdot e^{-2W} \quad (2)$$

where q_j and f_{Mj} are the magnetic interaction vector and the magnetic form factor for atom j , respectively. Therefore, similar to the diffraction patterns caused by the crystal structure of polycrystalline materials, the magnetic diffraction peaks arising from the magnetic structure are observed instead of canceling each other out as long as the diffraction conditions are met. In other words, neutron diffraction from different planes of atoms produces a diffraction pattern, which contains information about the atomic (magnetic moment) arrangement within the crystal.

Because the Mn_3AgN sample exhibits long-range magnetic order within each grain, i.e. a periodic magnetic structure, an additional contribution to the diffraction peaks due to the magnetic structure occurs.

We observed these peaks which were used to identify the magnitude and direction of the magnetic moments in the long-range magnetically ordered state. The following sentence has been added to the main text:

However, the refinement of the diffraction peaks reveals the magnitude and direction of the magnetic moments in the long-range magnetically ordered state.

In addition, we would like to emphasize that for isostructural Mn_3AgN no phase transitions are observed in the resistivities nor in the neutron scattering intensities (see Supplementary Information). Therefore, we have added the following sentence to the main text:

The strong link between the variation of the resistivities ρ and the variation of neutron diffraction intensities P_3 and P_4 across the phase transition is corroborated by comparison with the results for isostructural Mn_3AgN , where neither a variation of ρ nor a variation of intensities P_3 and P_4 with temperature is observed (Supplementary Information Figs. S5, S10).

The authors mentioned that effects from crystalline anisotropy are cancelled while effects from exchange interaction remain. Why are the effects from magnetic structures not cancelled in polycrystals? Are the effects under the magnetic field just from the remanence instead of different magnetic structures? How to distinguish the effects of the magnetic structure from the remanence?

Answer: That is a very good question. To answer it, we first note that the magnetic state of the collinear antiferro- and ferromagnets can be represented by a single vector whose orientation is determined by the competition between the crystalline anisotropy and the magnetic field. In a polycrystalline sample, the orientation of the averaged order parameter (magnetization or Néel vector) depends only on the orientation of the external field, assuming that all crystallographic orientations are equally represented. Similarly, the structure of the averaged resistivity tensor in the presence of the magnetic field corresponds to uniaxial symmetry with the symmetry defined by the magnetic field.

In contrast, the magnetic state of non-collinear antiferromagnets, one of which we are considering here, is represented by a multivector order parameter (which some researchers call an octupole). This order parameter describes not only the orientation of the magnetic vectors with respect to the crystallographic axes, but also the relative orientation of the magnetic sublattices. The angles between neighboring spins are determined by strong exchange interactions, the orientation with respect to the crystallographic axes is determined by the magnetic anisotropy, which is of spin-orbit origin and much weaker. The external magnetic field can modify the structure in two ways: by changing the relative angles between the spins, or by rotating the spin structure as a solid (keeping the relative angles fixed) with respect to the crystallographic axes. In the first case, the final magnetic structure depends on the absolute values of the field projection on the ordering plane rather than on the orientation of the magnetic field. In other words, the effect of the magnetic field is equivalent to uniaxial pressure. Averaging over the crystallographic orientations in this case reduces the value of the "effective pressure" but does not cancel it out.

In reality, however, the magnetic field produces both effects: it changes the angles between the spins and induces rotation of the spin ordering plane with respect to the crystallographic axes. Although the latter effect is small due to its relativistic nature, it reduces the symmetry of the magnetic structure to a trivial group, allowing the existence of non-diagonal components of the resistivity. However, the values of the non-diagonal components depend on the absolute value of the magnetic field and do not cancel out after averaging over the grains.

To address all these rather nontrivial issues, we have followed the advices of both Reviewers and

have added further comments on the magnetic structure and averaging in Note 5 of the Supplementary Information.

Regarding remanence: Remanence would be sensitive to the orientation of the magnetic field with respect to the sample, an effect we do not see. We also observe similar dependencies of transverse resistivity on magnetic field and temperature, suggesting a magnetic phase transition rather than a remanence effect.

We have modified the theory part to give more explanations, see our reply to your comment 2.

4. The theory model analyze the effects at a macroscopic level, which is usually used for single crystal and unproper for polycrystals. The arguments for polycrystals is too casual in Line 260-270. More calculation details should be given, and simulation for polycrystal and multi-domains should be performed.

Answer: Following the advice of both Reviewers we have added further comments on the magnetic structure and averaging in Note 5 of the Supplementary Information. It should be noted that the multidomain sample restores cubic symmetry, whereas the powder sample restores $O(3)$ symmetry (assuming an equiprobable distribution of elements). However, both symmetries are high enough to eliminate all crystallographic features. We therefore consider the model of the powder material that is relevant to our samples.

5. The M-H curve should be added to support the average magnetic moments (Line 159). Does the curve have a diamagnetic or paramagnetic background? Is the background deducted when calculating the average magnetic moments? What is the magnetic field the authors used to calculate the average magnetic moment of Mn atoms?

Answer: We thank the Reviewer for addressing this point. According to the Reviewer's comment we have revised Figure S1 in the SI which now includes the M - H curve for high (main panel) and low magnetic fields (inset).

In Figure S1a, the solid line indicates a linear behavior of $1/M \sim T$ with a Néel temperature $T_N = 411$ K, signifying that $Mn_3Zn_{0.5}Ge_{0.5}N$ is paramagnetic above T_N and magnetically ordered below. Figure S1c shows a linear field dependence of isothermal magnetization M - H which suggests that an antiferromagnetic state is observed at 2 K. We did not observe a diamagnetic background. Therefore, we did not apply any background correction: We obtain the average magnetic moment of Mn in $Mn_3Zn_{0.5}Ge_{0.5}N$ at $T = 2$ K from $M \sim 16$ A/m in 10 mT (Fig. S1b) and from the difference $\Delta M(H)$ of the remanent magnetization at $H=0$ (Fig. S1c, inset). This is now explained in the revised version of the Supplementary information

At zero field, a magnetization $M(0) = \Delta M(0)/2 = 1.5$ A m⁻¹ corresponding to 0.4×10^{-5} μ_B/Mn is verified by the isothermal magnetization curves at 2 K within our experimental accuracy of 0.3 A m⁻¹ (Fig. S1d). These tiny magnetizations agree with a very small NMR shift (Fig. S2) and the weak linear dependence on H confirms that the change of the magnetic structure below $T^ = 3.7$ K is very small.*

Figure S1 | **a**, Temperature dependence of inverse magnetization of $Mn_3Zn_{0.5}Ge_{0.5}N$. Solid line indicates a linear behavior. **b**, Temperature dependence of $M/\mu_0 H$ for various H . The short-dash lines indicate the linear behavior. **c**, Isothermal magnetization $M(H)$ at $T = 2$ K and 5 K. Inset shows $M(H)$ at 2 K in low fields where $\Delta M(0)$ is the difference of $M(0)$ between the up sweep and down sweep of the field H .

as well as in the revised main text:

The integral magnetization M of $Mn_3Zn_{0.5}Ge_{0.5}N$ is very small and does not show a transition at low temperatures at 2 K apart from a shallow increase below 5 K (Supplementary Fig. S1 b). From the temperature dependence of M we estimate a tiny average magnetic moment of $3.6 \times 10^{-5} \mu_B/Mn$ at $T = 2$ K in a weak magnetic field of 10 mT and $0.4 \times 10^{-5} \mu_B/Mn$ from the magnetization $M(0)$ in zero field (Supplementary Fig. S1).

6. Are the R-T curves measured during cooling process or heating process? Curves from both processes should be given to exclude the transition of the structure as a double check.

Answer: Yes, the R-T curves were measured during the cooling and heating processes and the behavior of these two processes was found to be consistent with negligible hysteresis. In Fig. 2 b of the revised manuscript we have included data for cooling and warming. We also added two panels to Fig. S3 a,b in the Supplementary Information which show $\rho_{\perp}(T)$ and $\rho_{\perp}(H)$ of sample #1 in response to comment 2 of Reviewer #2, see below. Fig. S3 b shows data for increasing and decreasing magnetic fields, indicated by arrows, with negligible hysteresis.

Figure S3 | **a**, Transverse resistivity $\rho_{\perp}(T)$ (sample #1) for various magnetic fields H applied perpendicularly to the current direction. **b**, $\rho_{\perp}(H)$ in perpendicular magnetic field H at various temperatures T . The transition field $H_0 = 30$ mT was determined from the resistivity at 1.8 K. The sweep direction of the magnetic field is indicated by arrows. **c**, Field dependence of the transverse resistance ratio R/R_0 for $\text{Mn}_3\text{Zn}_{0.5}\text{Ge}_{0.5}\text{N}$ (sample #2) at low temperatures, where R_0 is the value at $T = 4$ K in zero field and H_0 is 120 mT.

Reviewer #2 (Remarks to the Author)

Deng et al. reported a study on magnetic phase transitions in a non-collinear antiferromagnet (NCAFM) $\text{Mn}_3\text{Zn}_{0.5}\text{Ge}_{0.5}\text{N}$. The main method is longitudinal and transverse resistivity measurements under magnetic fields, where the boundaries of the R-H curve indicate the phase transition points. The experimental results are explained by a phenomenological model on the topological orbital momenta induced by magnetic fields.

NCAFM are attracting increasing interest due to their non-trivial spin and electronic structures. Particularly, many NCAFM exhibit multiple magnetic phases, whose magnetic and electronic properties are still unclear, posing challenges to advance the fundamental understanding and implementation of NCAFM spintronics. An in-depth study that determines the phase transition diagram and the characteristic magneto-transport effects, as done in this study, is of certain interest and importance, thus being potentially publishable in Nature Communications. However, the following comments should be addressed before I can recommend for publication.

1. Line 97: It has been reported that polycrystalline NCAFM Mn_3Sn exhibits a large AHE. I don't think the discussion that excludes "symmetry effects and AHE" from the origin of the transverse resistivity is convincing enough. More evidence or analysis should be provided.

Answer: We already mentioned on p. 5 & 6 of the manuscript that the magnetoresistance behavior cannot be explained by an AHE because (i) the magnetoresistances ρ_{\parallel} and ρ_{\perp} below T^* are even-order

(symmetric) functions of the magnetic field and (ii) the magnetoresistance contribution below T^* disappears by application of a magnetic field. We did not question the presence of an AHE in polycrystalline NCAFM like Mn_3Sn . To clarify this further we have revised the last paragraph of the introduction:

Here we report an investigation on A-site doped $Mn_3(Zn_{0.5}Ge_{0.5})N$ powder samples where spontaneous changes occur in the longitudinal and transverse resistivity at low temperatures which are susceptible to weak magnetic fields and are even-order (symmetric) functions of the applied magnetic field. We attribute the resistance changes to a magnetic phase transition between different non-collinear phases. (...) The unexpected and surprising result that in a polycrystal of non-collinear AFM order a finite transverse resistivity remains in zero magnetic field and vanishes in magnetic field is not due to symmetry effects or the AHE and provides information about the nontrivial magnetic phase diagram when properly investigated.

Regarding the size of the AHE in polycrystalline Mn_3Sn we have added two more references [32,33]:

This is smaller than the transverse conductivities of $150 - 500 \Omega^{-1}cm^{-1}$ arising from an AHE observed for other non-collinear antiferromagnets with triangular magnetic order like single-crystalline Mn_3Ge and Mn_3Sn ^{9,31} but similar to polycrystalline Mn_3Sn films^{32,33}.

2. Fig. 2: The longitudinal and transverse resistivity was taken from two samples. Is there a specific reason for not measuring in the same sample? How does the sample-to-sample variation influence the conclusion?

Answer: Initially we preferred to focus the discussion on a limited number of data. We now have added two more panels of $\rho_{\pm}(T)$ and $\rho_{\pm}(H)$ of sample #1 to Fig. S3 a,b in the Supplementary Information for completeness, please see our response to comment 6 of Reviewer #1 above. As described in our manuscript, several samples cut from the same batch exhibit an overall similar behavior, indicating that sample-to-sample variations are small and will not affect our conclusions.

3. Line 59: The authors attributed the MR effects in refs. 14 and 15 as large or even “colossal” AMR, whereas their MR is on the order of 0.1%. Please reconsider the terminology.

Answer: It was not our intention to claim a large AMR in our samples. The large AMR was mentioned in refs. 14 and 15. We removed “colossal” in the revised manuscript:

Large AMR values have been observed at magnetic-field induced transitions between different AFM phases¹⁵

4. Line 116: The transition field on the order of 10 to 100 mT is actually far below the expected robustness of an AFM. What is the reason?

Answer: We thank the Reviewer for this question. While in ferromagnets and collinear antiferromagnets the magnetic ordering can be interpreted on the basis of the Heisenberg exchange model with one or a few exchange constants, non-collinear antiferromagnets require a more complicated description. In particular, the stabilization of non-collinear structures can be explained by biquadratic exchange (or similar higher order spin-spin correlations such as $S_{1S_2S_3S_4}$). The role of the higher order term has been elucidated for e.g. $MnGe$ and $FeGe$ (see Ref. [42]) based on ab initio calculations. In addition, if both types of exchange constants are of the same order, the material can exhibit phase transitions between phases with different spin angles (see Ref. [44] for a phenomenological classification of the phases). In the case of $Mn_3Zn_{0.5}Ga_{0.5}N$, the Heisenberg exchange is temperature dependent and

favors coplanar Γ^{5g} antiferromagnetic ordering over a wide range of temperatures from 400 K down to 4 K. However, we believe that below 4 K the Heisenberg exchange constant changes sign and the noncolinear ordering is stabilized by the biquadratic exchange. In this temperature region, close to the phase transition, the system becomes soft and thus sensitive to much lower fields compared to the region around 300 K.

[42] Grytsiuk, S. et al. Topological–chiral magnetic interactions driven by emergent orbital magnetism, *Nature Comm.* **11**, 511 (2020).

[44] Gomonaj, E. V. & L'vov, V. A. Phenomenologic study of phase transitions in noncollinear antiferromagnet of metallic perovskite type. *Phase Transitions* **38**, 15-31 (1992).

To address this issue, we added the following phrase to the main text (Line 121):

This extraordinary behavior is unexpected given the usual robustness of AFM order against moderate magnetic fields. In $Mn_3Zn_{0.5}Ga_{0.5}N$ it occurs only close to the phase transition between different non-collinear spin textures, where the Heisenberg exchange is small and the structure is stabilised by the higher order terms of the exchange nature (e.g. biquadratic exchange).

5. Line 137: It is surprising that the transverse resistivity does not depend on the magnetic field orientation, which is different from the polycrystalline FM samples. The authors should explain more explicitly on this independence, rather than simply ascribe it to the polycrystalline structure of the AFM.

Answer: To address this issue we have rewritten the first part of the section “Theoretical model” (lines 222-238), please see also our answer to comment 1 of Reviewer #1.

6. Following comment 5, does the longitudinal resistivity show any angle dependence?

Answer: It does not and should not. However, the transverse component is a much more sensitive tool for detecting the different phases because it appears only in the low-symmetry phase and has a purely magnetic origin. In contrast, the magnetic contribution to the longitudinal component is $\sim 1\%$ (see Fig. 2a,b). We have now added $\rho_{||}(H, \varphi)$ for sample #2 as panel c to Fig. S4 of the Supplementary Information as an example..

Figure S4 | **a**, Angular dependence of $\rho_{\perp}(H)$ of $Mn_3Zn_{0.5}Ge_{0.5}N$ for various angles ω between the directions of field H and voltage V_{\perp} at 2 K where $H_0 = 120$ mT. ω is the angle of in-plane rotation around the surface normal. **b**, Angular dependence of $\rho_{\perp}(H)$ for various out-of-plane angles φ between the directions of field H and current I at 1.8 K. **c**, Angular dependence of $\rho_{||}(H)$ for various out-of-plane angles φ between the directions of field H and current I at 1.8 K.

7. Line 242: When a magnetic field H is applied to the NCAFM, M is replaced by χ^*H , while N_1 and N_2 remain unchanged. Will N_1 and N_2 also be functions of H ?

Answer: It is true that the length of the Néel vectors also changes with the magnetic field. However, the corresponding corrections to the energy will be of the order of $\chi^4 H^4$. The maximum field value used in the experiment (120 mT) produces a magnetic moment of 100 A/m (see Fig. S1), which is still much smaller than the saturation magnetization of $\sim 10^5$ A/m. We have therefore neglected these corrections. To clarify this point, we have added the following sentences to Supplementary Note 5:

Here we have neglected the field-induced variation of the Néel vectors \mathbf{N}_1 and \mathbf{N}_2 , assuming that the field-induced component of the magnetisation is small, $M \ll M_s$. This assumption is relevant for the small field values used in the experiment (see Fig. S1).

Condition (ii) defines additional relations between the vectors: $\mathbf{N}_1^2 + \mathbf{N}_2^2 + \mathbf{M}^2 = 3M_s^2$, $2\sqrt{2}(\mathbf{N}_1\mathbf{M}) = \mathbf{N}_1^2 - \mathbf{N}_2^2$, $\sqrt{2}(\mathbf{N}_2\mathbf{M}) = (\mathbf{N}_1\mathbf{N}_2)$.

8. The authors mentioned the uniqueness of using polycrystalline AFM materials for spintronic applications. I suggest the authors to enrich the discussions by referring to the relevant content and references in a recent review Nat. Mater. 22, 684–695 (2023).

Answer: We thank the Reviewer for this suggestion. We have added two sentences mentioning the advantages of polycrystalline materials for applications and two additional references by Han et al. [8] and DuttaGupta et al. [17]:

In particular, antiferromagnets with non-collinear magnetic structure are quantum materials that exhibit unique spin-dependent properties^{7,8},

From a materials science point of view, polycrystalline materials are just as interesting as single crystals because they are more widely used in applications, they are compatible with Si-based electronics⁸, and crystallographic dependencies are balanced out by the randomly arranged crystallites. In this context, the potential of polycrystalline metals for AFM spintronics has been demonstrated by successful spin-orbit torque switching of polycrystalline AFM heterostructures¹⁷.

REVIEWERS' COMMENTS

Reviewer #1 (Remarks to the Author):

This work investigated that polycrystalline $\text{Mn}_3\text{Zn}_{0.5}\text{Ge}_{0.5}\text{N}$ with non-collinear antiferromagnetic order show subtle transitions between magnetic phases of different symmetry. The revision includes some new experimental data on magnetization, transverse resistivity, and its angular dependence. Some writing problem has also been revised. From my side, I think the present version is up to the standard for the publication.

Reviewer #2 (Remarks to the Author):

The authors have addressed my comments in a satisfactory manner and revised the manuscript accordingly. I am happy to support the publication in Nature Communications.